# Bidirectional Patellar Luxation in Small- or Miniature-Breed Dogs in Japan; Patient Characteristics and Radiographic Measures Compared with Medial Patellar Luxation

**DOI:** 10.3390/vetsci10120692

**Published:** 2023-12-05

**Authors:** Itaru Mizutani, Reo Nishi, Masahiro Murakami

**Affiliations:** 1Mori Animal Hospital, Suzuka-shi 513-0806, Japan; contact@mori-vh.com; 2Department of Veterinary Clinical Sciences, College of Veterinary Medicine, Purdue University, West Lafayette, IN 47907, USA; rnishi@purdue.edu; 3Department of Veterinary Clinical Pathobiology, Graduate School of Agricultural and Life Sciences, The University of Tokyo, Bunkyou-ku, Tokyo 113-8657, Japan

**Keywords:** orthopedic disease, patellar luxation, small-breed dogs, miniature-breed dogs, trochleoplasty, tibial tuberosity transposition

## Abstract

**Simple Summary:**

Bidirectional patellar luxation (BPL) is a not-widely described form of patellar luxation. The purpose of this study was to describe common breeds and radiographic measures associated with BPL in dogs in Japan, compared with dogs with medial patellar luxation (MPL). The most common breed with BPL in the present study was Toy Poodles (odds ratio: 7.05), and patella alta in the extended-stifle position was more common in the BPL group (23.4%) than in the MPL group (0.8%). However, there were no significant differences in radiographic-patellar-ligament length indices between the BPL and MPL groups. The study highlights the common breeds with BPL, particularly Toy Poodles, and suggests that the occurrence of BPL may be related to stifle lengthening. This information will contribute to a better understanding of this unique form of patellar luxation, and may help guide more patient-specific treatment options.

**Abstract:**

Bidirectional patellar luxation (BPL) is a relatively rare form of patellar luxation, with limited information reported regarding breed predisposition and etiology. The purpose of this study was to describe the patient characteristics and radiographic measures of proximodistal patellar position associated with BPL in dogs in Japan, compared with dogs with medial patellar luxation (MPL). A retrospective medical record search of surgically corrected MPL and BPL dogs was performed, and breed, age, sex, body weight, and presence of the patella alta in the extended-stifle position were recorded. The ratio of the patellar ligament length to patella length (PLL/PL) and the ratio of the distance between the proximal pole of the patella and the femoral condyle to patella length (A/PL) were measured on stifle radiographs. A total of 35 dogs with BPL and 95 dogs with MPL were included. The BPL was most commonly present in Toy Poodles (odds ratio compared to MPL dogs: 7.05) in the present study. There were no significant differences in age, sex, or body weight between the BPL and MPL groups. Patella alta in the extended-stifle position was more common in the BPL group (23.4%) than in the MPL group (0.8%). However, there were no significant differences in PLL/PL or A/PL between the BPL and MPL groups. The study highlights BPL in different dog breeds in Japan, and suggests that the occurrence of BPL may be related to stifle extension. However, more research is needed to fully understand the etiology of BPL.

## 1. Introduction

Patellar luxation is a common orthopedic abnormality in small- and miniature-dog breeds, with medial, lateral, and bidirectional types. Bidirectional patellar luxation (BPL) is a relatively rare form of patellar luxation, and was not observed in a previous publication evaluating the frequency of medial and lateral patellar luxation in dogs [1,2]. However, BPL has been reported in several countries, accounting for 6.5% of all patellar luxation cases in small- and miniature-dog breeds in Austria [3], and 3% [4] and 7% [5] of Pomeranians [4] and Kooiker dogs [5], respectively, in Thailand, and 2.4% in Sweden [6]. Although the authors have observed BPL more frequently in Toy Poodles in Japan, to date, to the best of the author’s knowledge, only one publication in Japanese has been published describing the incidence of BPL in Toy Poodles in Japan [7]. No previous studies have investigated common breeds with BPL in small or miniature breeds in Japan.

Recognition of bidirectional patellar luxation as a distinct clinical entity is crucial, due to its specific management protocols [8]. It is essential not only to diagnose BPL, but also to understand the differences in predisposing anatomical features, particularly regarding proximodistal patellar position, to better comprehend effective treatment strategies.

Excluding trauma or injury as a cause, congenital or developmental anatomical features are believed to be associated with the pathogenesis of patellar luxation in dogs. These features include a shallow or absent femoral groove, abnormal alignment of the knee and leg, skeletal abnormalities, or hormonal influences during growth, particularly in small-breed dogs with medial patellar luxation (MPL) [9,10,11]. Although the cause of patellar luxation is considered multifactorial, the proximodistal patellar position has been discussed as one of the possible factors affecting the development of patellar luxation in dogs [12,13,14,15,16]. A proximally positioned patella (patella alta) has been reported to be associated with patellar instability in humans [17,18,19,20], as the patella is positioned proximal to the femoral trochlea and loses support from the trochlear ridge. 

The patellar ligament length (PLL) to patellar length (PL) ratio (PLL/PL), and the distance between the proximal pole of the patella and femoral condyle (A) in relation to the PL ratio (A/PL) have been used for evaluating proximal patellar position in dogs [15,16,21,22]. The stifle with a PLL/PL value > 1.97–2.06 or an A/PL value > 2.03 was considered patella alta [15,16]. In cases of BPL, affected dogs typically exhibit knee joint overextension while standing [8]. During this overextension, the patella is positioned proximally in the trochlear groove, resulting in reduced support from both the medial and lateral trochlear ridges. However, the PLL/PL was not significantly different among seven Pomeranian with BPL and Pomeranian with normal stifle [8]. There is no study evaluating PLL/PL or A/PL in larger numbers of dogs with BPL in multiple small- and miniature-dog breeds.

The primary objective of this study is to describe the population characteristics, including breed, in dogs with BPL in Japan, and compare them with the population characteristics observed in dogs with MPL. The secondary objective is to measure the PLL/PL and A/PL values from radiographs of small- and miniature-dog breeds and compare them between MPL and BPL groups, aiming to identify factors associated with BPL in these dogs. Based on preliminary observations, we hypothesize that the Toy Poodle is the breed most commonly diagnosed with BPL in Japan, and that there is a discrepancy in breed distribution when compared with conventional MPL. We also hypothesize that the PLL/PL and A/PL will not differ significantly between the BPL and the MPL groups, similar to the previous publication in Pomeranian [8].

## 2. Materials and Methods

### 2.1. Case Selection and Medical Record Review

This study was a single-center, retrospective, observational, and descriptive analysis conducted by reviewing the medical records of a primary veterinary hospital in Japan. Medical records were reviewed for a thirteen-year period from 2010 to 2023 to identify dogs diagnosed with MPL or BPL, who subsequently underwent surgical correction. Cases were included or excluded based on the decision of a veterinarian with 20 years of experience in orthopedic surgery (I.M.) and an American College of Veterinary Radiology (ACVR) board-certified radiologist (M.M.). Permission to use the medical records for this study was obtained from both the hospital owner/director and the owners of the animals. Data extracted from the medical records for each dog included breed and sex. Age and body weight of dogs were recorded for each stifle at the time of surgery for MPL or BPL.

The presence of MPL or BPL and the presence of patella alta in the extended-stifle position were recorded from the medical records for each stifle of the dogs.

Patellar luxation assessment was conducted during a physical examination without sedation, following previously established criteria [9,23,24]. The types of patellar luxation were confirmed under general anesthesia immediately before surgery in all dogs. Patella alta in the extended-stifle position was assessed during surgery. The condition was diagnosed when the center of the patella was positioned proximal to the most proximal border of the femoral groove/trochlear ridge in a fully extended stifle, under general anesthesia. Unilateral or bilateral patellar luxation was recorded. The dogs with unilateral MPL or bilateral MPL were assigned to the MPL group, and the dogs with at least one stifle with BPL (unilateral BPL, mixed bilateral MPL and BPL, and bilateral BPL) were assigned to the BPL group for statistical analysis.

### 2.2. Radiographic Measurement

Further radiographic assessment was conducted for each stifle in the study. Pre-operative radiographs were collected and reviewed by the authors (I.M. and M.M.). Cases lacking both craniocaudal and mediolateral pre-operative radiographic views were excluded. Radiographs were deemed satisfactory, based on the following criteria: superimposition of the femoral condyles, inclusion of the metaphyseal–diaphyseal junction of both the femur and tibia in the mediolateral view, and symmetrical bisection of the femoral cortices by both fabellae in the craniocaudal view. Cases whose radiographs were unsatisfactory in terms of positioning were excluded. Cases with a history of stifle joint surgery or significant degenerative joint disease that would preclude accurate measurements were also excluded from the study.

All measurements were performed by a single investigator (I.M.), who underwent training in taking appropriate radiographic measurements before the study began. Measurements were conducted on standard mediolateral and craniocaudal radiographic projections of the stifle using open-source DICOM viewing software (Horos v3.3.6, HorosProject™).

As the PLL is known to be influenced by the stifle joint angle [15], we first measured the stifle joint angle in mediolateral views of the stifle, using a previously described method for small-breed dogs [12]. The stifle angle was defined as the caudal angle formed by the anatomical axes of the distal femur and proximal tibia. The extension of the line connecting the two centers of the femoral width was defined as the distal femoral anatomical axis. The distal femoral width was one femoral condyle length from the proximal end of the trochlea, and the proximal width was half the length of the femoral condyle from the distal one. The proximal tibial anatomical axis was defined as the extension of the line connecting the center of the tibial width, which was 1.5 times the length of the proximal tibial width from the tibial plateau and the cranial notch to the tibial plateau (Figure 1A). Cases with stifle angles outside the range of 70 to 110 degrees [15] were excluded from further analysis.

Next, we measured the PL, PLL, and vertical line (A), according to previously published methods [15]. PL was defined as the distance from the most-proximal to the most-distal aspect of the patella, while PLL was the distance from the most-distal aspect of the patella to the tibial tuberosity (Figure 1B). Care was taken to exclude osteophytes when noted at the distal aspect of the patella. The PLL/PL ratio was then calculated. The distance from the proximal border of the patella to the level of the femoral condylar articular margin (A) was measured on a craniocaudal radiograph (Figure 1C) and indexed to the PL to provide A/PL. The value for PL was measured on the corresponding mediolateral view of the stifle because superimposition over the distal aspect of the patella prevented adequate observation of the distal pole of the patella on caudocranial and craniocaudal projections.

### 2.3. Statistical Analyses

The software used for the analyses was free, open-source statistical software (EZR version 1.61, Jichi Medical University, Saitama, Japan; graphical user interface for R, the R Foundation for Statistical Computing, Vienna, Austria).

The odds of occurrence of BPL relative to MPL in each breed were evaluated using a two-tailed Fisher’s exact test. Results were expressed as odds ratios and 95% confidence intervals (CIs). Odds ratios and 95% CIs were calculated for each breed, to compare the occurrence of BPL to MPL. Mean (±standard deviation; SD) body weight and age between BPL and MPL groups were evaluated using a two-tailed Man–Whitney U-test, due to the result of the Shapiro–Wilk normality test. To evaluate the association between the presence of patella alta in the extended-stifle position and the type of patellar luxation (MPL or BPL), a two-tailed Fisher’s exact test was performed, and the results were expressed as odds ratios comparing the occurrence of patella alta in BPL to MPL, with their 95% CIs. Comparisons of PLL/PL and A/PL between the BPL and MPL groups were evaluated using either a two-tailed unpaired Student’s *t*-test or a two-tailed Mann–Whitney U-test, depending on the results of the Shapiro–Wilk normality test and F-test. A *p*-value less than 0.05 was considered statistically significant.

## 3. Results

In this study, we examined a total of 130 dogs, which were divided into two groups: 95 dogs in the MPL group and 35 dogs in the BPL group. Among these dogs, 42 had bilateral patellar luxation, while 88 had unilateral patellar luxation. The bilateral-patellar-luxation subgroup consisted of 30 dogs with bilateral MPL, 8 dogs with MPL and contralateral BPL, and 12 dogs with bilateral BPL (Table 1).

The distribution of sex and neutering status within each subgroup is described in Table 2.

Dogs were assigned to either the MPL or BPL group, based on their patellar luxation type. Dogs with unilateral MPL or bilateral MPL were assigned to the MPL group, while dogs with at least one limb exhibiting BPL (unilateral BPL, unilateral BPL and contralateral MPL, or bilateral BPL) were assigned to the BPL group.

The incidence of BPL relative to MPL across various breeds, along with the corresponding odds ratios and 95% Cis, are described in Table 3. The Toy Poodle emerged as the most commonly diagnosed breed in the BPL group, representing 53.5% of BPL dogs. Subsequent breeds in descending order of the percent presence of BPL in dogs with MPL or BPL included the Cavalier King Charles Spaniel (50.0%), Pomeranian (33.3%), Mixed breed (27.8%), Yorkshire Terrier (11.1%), and Chihuahua (9.1%). When comparing the likelihood of developing BPL in at least one limb in the group of dogs with either MPL or BPL, Toy Poodles exhibited an odds ratio of 7.05 (95% CI: 2.82–18.6, *p* < 0.001), while Chihuahua had an odds ratio of 0.21 (95% CI: 0.04–0.74, *p* < 0.001) (Table 3). This indicates that in a group of dogs with either MPL or BPL, the incidence of BPL in Toy Poodles is 7.05 times more common than in other breeds, and the incidence of BPL in Chihuahuas is 0.21 times less common than in other breeds.

The mean body weight (±SD) and range for dogs with MPL and BPL were as follows: the MPL group had a mean weight of 4.01 kg (±2.43) and a range of 1.7–15 kg, while the BPL group had a mean weight of 3.34 kg (±1.26) and a range of 1.3–7.0 kg. No significant difference in body weight at the time of surgery was observed between the MPL and BPL groups (*p* = 0.37).

Similarly, the mean age (±SD) and range for dogs with MPL and BPL were examined. The MPL group had a mean age of 2.6 years (±2.4) and a range of 0.5–12 years, while the BPL group had a mean age of 2.2 years (±2.2) and a range of 0.7–10 years. No significant difference in age at the time of surgery was observed between the MPL and BPL groups (*p* = 0.50).

The analysis that follows from this point was performed for each stifle, with a total of 180 stifles included in the study. The presence of patella alta in the extended-stifle position was observed in 0.8% of the MPL stifle and 23.4% of the BPL stifle (Table 4). There was a significant difference between the presence of patella alta in the extended-stifle position in different types of patellar luxation (MPL or BPL) (odds ratio: 39.4, 95% CI: 5.41–1731.8, *p* < 0.001).

Following the exclusion of unsatisfactory radiographs, we analyzed a total of 124 stifle joints, using the craniocaudal and mediolateral views. We measured the stifle joint angles for all 124 mediolateral views. Subsequently, we excluded 59 stifles, due to their joint angles falling outside the previously reported range [15]. After these exclusions, 65 stifles were included for further analysis, which included 48 stifles with MPL and 17 with BPL.

The mean PLL/PL ratio for the MPL group was 2.03 ± 0.18, and for the BPL group it was 2.09 ± 0.20 (Table 5). There was no significant difference in the PLL/PL ratio between the two groups (*p* = 0.32). The mean A/PL for the MPL group was 1.93 ± 0.19, and for the BPL group it was 1.95 ± 0.25 (Table 5). There was no significant difference in the A/PL ratio between the two groups (*p* = 0.69).

PLL/PL and A/PL in each common breed of dogs with MPL and BPL were also evaluated, and the mean (±SD) and *p*-value were calculated. Calculations for breeds other than Toy Poodle and Mixed breed were not performed because there were not enough numbers in one or both groups to perform statistical analysis. There was no significant difference in PLL/PL or A/PL between MPL and BPL in any common breed of dog.

In the present study, we also evaluated the ratios of PLL/PL and A/PL within the two breed groups with a sufficient number of observations (Toy Poodle and Mixed breed), but found no significant differences.

## 4. Discussion

Our study aimed to describe anatomical features associated with BPL in small- and miniature-dog breeds in Japan, in comparison with MPL. We found that the Toy Poodle breed was the most commonly diagnosed with BPL, aligning with our initial hypothesis. Furthermore, the presence of BPL compared to MPL was higher in our study than generally reported, which could be due to the popularity of the Toy Poodle breed in Japan. We found no significant difference in the proximodistal patellar position (PLL/PL and A/PL values) between the BPL and MPL groups, although the presence of patella alta in the extended-stifle position was significantly present in BPL stifles, which was also consistent with our hypothesis.

MPL is a common orthopedic problem in dogs, especially in young dogs and small breeds [1,25,26]. Previous studies have reported the median or mean age of dogs with MPL to be approximately 3 to 4 years, with a wide range of ages, from less than 1 year to over 10 years [15,25]. The age of dogs with BPL is limited, but one study of Pomeranian dogs with BPL reported a mean age (±SD) of 35.4 (±29.7) months [8], and another study reported a median age of 1.5 years [6]. In our study, the mean age (±SD) at the time of surgery for dogs with MPL and BPL was 2.71 (±2.31) and 1.98 (±2.22) years, respectively, which is consistent with the age distribution reported in previous studies of MPL and BPL dogs. We also found no significant age difference between the MPL and BPL groups.

MPL is significantly more common in small-breed dogs, with the incidence of MPL being 12 times higher in small breeds compared to large breeds [26]. The reported mean body weight of dogs with MPL varies, with one report indicating a mean adult-dog weight of 8.0 kg, ranging from 1.4 to 65.5 kg [25]. One study comparing stifle radiographs in dogs with MPL with dogs without stifle disease reported a mean body weight (±SD) of 28.1 (±9.8) kg in dogs with MPL, with no significant difference in body weight between dogs with MPL and dogs without stifle disease [15]. The literature on the body weight (±SD) of dogs with BPL is limited, with one study reporting a mean weight of 2.8 (±0.5) kg, although this study included only Pomeranians [8]. No previous studies have compared the body weights of dogs with MPL and BPL. In our study, we found no significant difference in body weight between the MPL and BPL groups (*p* = 0.89). Both groups had a low mean body weight (±SD): 3.14 (±0.98) kg for the MPL group and 3.12 (±1.13) kg for the BPL group. This trend toward lighter body weights is likely due to the breed predisposition of MPL and BPL in small-breed dogs, as well as the breed distribution in Japan, which has a large population of small-breed dogs.

BPL has been observed in several breeds worldwide [1,3,4,5,8], but our study shows a notable presence of BPL in Toy Poodles, which accounted for 53.5% of the BPL group. This prevalence could be due to the actual breed predisposition to BPL, the popularity of Toy Poodles in Japan [27,28] (the Toy poodle was most common breed in Japan in the study period (2010–2023), as reported by Japan Kennel Club [29,30]), or a combination of both. The second most-common breed with BPL in our study was the Cavalier King Charles Spaniel (CKCS). However, only two CKCSs were involved in the study, making it difficult to draw definitive conclusions about their predisposition to BPL. The third most-predisposed breed was the Pomeranian (33.3%), which is a breed reported with BPL in the previous publications [4,8].

Although there are only limited publications describing BPL, and BPL is considered a relatively rare form of patellar luxation [1,3,4,5,8], we found a relatively high number of dogs with BPL in our study, with 35 of the 130 dogs (26.9%) exhibiting BPL. Because small dogs, which are predisposed to MPL, are common in Japan, the dogs in our study were small (mean body weight of 4.01 kg in the MPL and 3.34 kg in the BPL group). Toy poodles are currently one of the most popular dog breeds in Japan; along with their possible predisposition to BPL, this is likely to have influenced the relatively high number of dogs with BPL observed in the current study.

In both the MPL and BPL groups, nearly one-third of the dogs had bilateral patellar luxation (31.6% in MPL and 34.3% in BPL), which aligns with the understanding that anatomical features strongly influence the occurrence of patellar luxation. This similarity between the MPL and BPL groups suggests a shared pathophysiology related to bilateral anatomical predisposition.

The condition known as patella alta is characterized by a patella that is more proximal than normal, in relation to the femur. This results in a loss of lateral support from the trochlear ridges and may be associated with patellar luxation [15,16]. The angle of the stifle is known to influence the occurrence of patella alta [15], with relatively proximal positioning of the patella occurring in conjunction with stifle extension. To objectively quantify patella alta, the PLL/PL or A/PL ratio is commonly assessed using stifle radiographs with a stifle angle between 70 and 110 degrees [15]. A PLL/PL ratio greater than 2.06 or an A/PL ratio greater than 2.03 is indicative of patella alta in medium to large breed dogs, according to a previous study [15].

Previous research has shown an association between MPL and patella alta [15]. The relationship between BPL and patella alta has also been studied in Pomeranians [8]. In Pomeranians diagnosed with BPL, the PLL/PL ratio was not significantly different from that of a healthy stifle [8]. This led to the conclusion that there is no evidence that anatomical patella alta is associated with BPL. In our current study, patella alta in extended-stifle position was significantly more common in the BPL group (23.4%), compared to the MPL group (0.8%). However, the objective measures of proximodistal patellar position (PLL/PL and A/PL ratios) did not show a significant difference between the MPL and BPL groups.

Our findings suggest a high prevalence of patella alta when the stifle is extended, but no objective measures of patella alta (PLL/PL and A/PL ratios) were observed in normally flexed stifles. This suggests that stifle extension resulting in a functional patella alta may contribute to the development of BPL, a previously proposed hypothesis [8]. This could potentially explain the common posture observed in Toy Poodles, particularly those with BPL, who often stand with an overextended stifle during physical examinations and while walking.

Therefore, further research is needed to evaluate the stifle angle during gait in Toy Poodles and compare it to other breeds. Such studies may provide valuable insight into the causes of functional patella alta in this breed.

Our study has a few limitations that should be noted. It is based on a relatively small sample size and is restricted to Japan, which has a unique distribution of small-breed dogs. The high prevalence of Toy Poodles in the country could have influenced our results. However, the Chihuahua, another common breed in Japan, showed a much lower presence of BPL (9.1%), resulting in a stark contrast in odds ratios for the presence of BPL compared to MPL between Toy Poodles (7.05) and Chihuahuas (0.21). Our study did not include large-breed dogs, which limits our ability to evaluate the prevalence of BPL in these dog breeds. However, MPL is rare in large-breed dogs, suggesting that the likelihood of BPL in large-breed dogs would be even less likely. Additionally, as our study was retrospective and included only cases with MPL or BPL that underwent surgical correction, minor MPL cases might not have been included. As such, we were unable to evaluate the occurrence of BPL in minor-degree-MPL dogs. Furthermore, the inconsistency in radiograph positioning, due to the retrospective nature of the study, might have influenced our results. Future prospective studies should aim for consistently and strictly positioned stifles at a fixed angle, for more accurate evaluation.

## 5. Conclusions

In conclusion, our investigation provides important insights into anatomical features and breed characteristics associated with BPL in small- and miniature-dog breeds in Japan. The Toy Poodle was the most common breed to be surgically treated for BPL in this study, and the occurrence of BPL may be related to stifle extension, as patella alta in the extended-stifle position was frequently observed in cases with stifle extension, although no differences in PLL/PL or A/PL ratios were found between MPL and BPL dogs. We believe that our findings will improve the understanding of the pathophysiology of BPL, and aid in the development of more effective diagnostic and therapeutic approaches for this condition.

## Figures and Tables

**Figure 1 vetsci-10-00692-f001:**
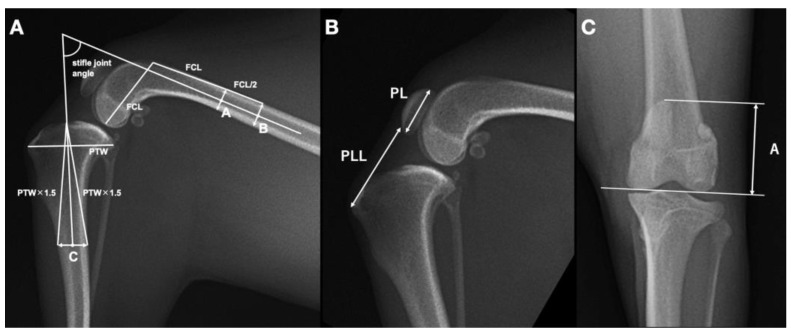
Radiographic assessment of stifle joint angle and proximodistal patella position in a dog using mediolateral (**A**,**B**) and craniocaudal (**C**) stifle radiographs. A: The stifle joint angle (white arc) was measured as the caudal angle between the anatomical axes of the femur and tibia. The femoral anatomical axis was determined by extending the line connecting points A and B, where point A is the midpoint of the femoral width at one femoral condyle length (FCL) proximal to the proximal end of the trochlea, and point B is half the FCL proximal to point A. The tibial anatomical axis was defined by extending the line connecting point C and the notch at the front of the tibial plateau, with point C being the midpoint of the tibial width at a level 1.5 times the length of the proximal tibial width (PTW) distal to the tibial plateau notch. B: The ratio of proximodistal patellar length (PL) to patellar ligament length (PLL) (PL/PLL) was calculated, where PL was the distance from the most-proximal to the most-distal aspect of the patella, and PLL was the distance from the most-distal aspect of the patella to the tibial tuberosity. C: The distance from the proximal border of the patella to the level of the femoral condylar articular margin (A) to PL ratio (A/PL), where PL was measured on the mediolateral view of the stifle.

**Table 1 vetsci-10-00692-t001:** The number of dogs with MPL or BPL. MPL, medial patellar luxation; BPL, bidirectional patellar luxation.

	MPL			BPL		
Breed	Unilateral	Bilateral	Total	Unilateral	Bilateral	Total
Toy Poodle	18	2	20	14	9	23
Chihuahua	21	9	30	3	0	3
Mixed breed	6	7	13	2	3	5
Yorkshire Terrier	5	3	8	1	0	1
Pomeranian	2	2	4	2	0	2
Cavalier King Charles Spaniel	0	1	1	1	0	1
Maltese	3	3	6	0	0	0
Shibainu	6	1	7	0	0	0
Papillon	1	1	2	0	0	0
Miniature Pinscher	0	1	1	0	0	0
Pekingese	1	0	1	0	0	0
Pug	1	0	1	0	0	0
Spitz	1	0	1	0	0	0
Total	65	30	95	23	12	35

**Table 2 vetsci-10-00692-t002:** The distribution of sex and neutering status within each subgroup (n). MPL, medial patellar luxation; BPL, bidirectional patellar luxation.

		Intact Males	Castrated Males	Intact Females	Spayed Females
MPL	Unilateral	25	4	27	9
	Bilatelaral	14	1	11	4
BPL	Unilateral	10	0	4	1
	Bilatelaral	7	0	2	3
MPL and Contralateral BPL		3	1	4	0

**Table 3 vetsci-10-00692-t003:** Presence of bidirectional patellar luxation (BPL) compared with MPL in dogs in Japan. This table presents the odds ratios for the presence of BPL compared with MPL in various dog breeds, analyzed using a two-tailed Fisher’s exact test. Breeds with a small number of cases were combined into the group “other breeds”, for statistical analysis. MPL, medial patellar luxation; BPL, bidirectional patellar luxation; CI, confidence interval.

Breed	Total	The Number of Dogs with MPL	The Number of Dogs with BPL	Occurrence of BPL (%)	Odds Ratio	Lower 95% CI	Upper 95% CI	*p* Value
Toy Poodle	43	20	23	53.5	7.05	2.82	18.6	<0.001
Chihuahua	33	30	3	9.1	0.21	0.04	0.74	<0.001
Other breeds	54	45	9	16.7	0.39	0.14	0.97	<0.05

**Table 4 vetsci-10-00692-t004:** Presence of patella alta in extended-leg position in canine stifles with MPL and BPL. Odds ratio was calculated using a two-tailed Fisher’s exact test.

	Patella Alta (−)	Patella Alta (+)	Presence of Patella Alta (%)	Odds Ratio	Lower 95% CI	Upper 95% CI	*p*-Value
MPL	132	1	0.8				
BPL	36	11	23.4	39.4	5.41	1731.8	<0.001

MPL, medial patellar luxation; BPL, bidirectional patellar luxation; CI, confidence interval.

**Table 5 vetsci-10-00692-t005:** Radiographic measurements of proximodistal patella position in canine stifles with MPL and BPL. Values are expressed as means ± standard errors. Differences between groups were analyzed using Student’s *t*-test.

	MPL	BPL	*p*-Value
PLL/PL	2.03 ± 0.18	2.09 ± 0.20	0.32
A/PL	1.93 ± 0.19	1.95 ± 0.25	0.69

PLL/PL, ratio of patellar ligament length to patellar length; A/PL, ratio of femoral condyle to patella length; MPL, medial patellar luxation; BPL, bidirectional patellar luxation.

## Data Availability

The data that support the findings of this study are available from the corresponding author upon reasonable request.

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
