# Peer review of "Bidirectional Patellar Luxation in Small- or Miniature-Breed Dogs in Japan; Patient Characteristics and Radiographic Measures Compared with Medial Patellar Luxation"

_vetsci, 2023, doi:10.3390/vetsci10120692_

Round 1

Reviewer 1 Report

Comments and Suggestions for Authors

The use of “stifle weight” and “stifle age” throughout the Methods and Results should be rephrased. For example, Line 89-90: “Age and body weight were recorded for each stifle” – stifle should be replaced with dog. The weights and ages refer to the dog not each individual stifle. The phrasing used for weights and ages in the Discussion should be used.

A few minor edits:

Line 64 and 65: Two sentences in a row start with “However” confusing the intended subject of the sentences.

Line 130: Reference needs to be filled in.

Lines 181-183: Correct singular vs plural female(s) and male(s).

Table 1: Check consistency with headings in bold. If the bold indicates an important feature, it should be explained in the legend  

Comments on the Quality of English Language

Overall good writing quality other than the "stifle" vs "dog" comment mentioned above.

Author Response

November 13, 2023

Subject: Revision and resubmission of manuscript ID vetsci-2703047to the MDPI Veterinary Sciences.

I have included the reviewer comments immediately after this letter and responded to them individually, indicating exactly how we addressed each concern or problem and describing the changes we have made. The revisions have been approved by all authors. The changes are highlighted by using the track changes mode in MS Word and using bold in the manuscript, and revised manuscript will resubmit with this letter.

Our response follows (the reviewer’s comments are in italics).

Reviewer: 1

Comments to the Author:
The use of “stifle weight” and “stifle age” throughout the Methods and Results should be rephrased. For example, Line 89-90: “Age and body weight were recorded for each stifle” – stifle should be replaced with dog. The weights and ages refer to the dog not each individual stifle. The phrasing used for weights and ages in the Discussion should be used.

Reply:

Thank you for your constructive feedback regarding the terminology used in the Methods and Results sections. We recognize that the terms "stifle weight" and "stifle age" could lead to confusion, as they inadvertently suggest that these measurements apply to the stifle itself rather than to the dogs. To clarify this point, we have carefully reviewed the manuscript and amended these terms to "age of the dog" and "body weight of the dog" wherever they appear. This revision aligns with the more accurate phrasing employed in the Discussion section and ensures consistency and clarity throughout the paper. We appreciate your attention to detail and agree that this change enhances the manuscript's precision.

Line 107-108:
Age and body weight of dogs were recorded for each stifle at the time of surgery for MPL or BPL.
Line 289-298:
The mean body weight (±SD) and range for dogs with MPL and BPL were as follows: MPL group had a mean weight of 4.01 kg (±2.43.) and a range of 1.7 – 15 kg, while BPL group had a mean weight of 3.34 kg (±1.26) and a range of 1.3 – 7.0 kg. No significant difference in body weight at the time of surgery was observed between MPL and BPL groups (P = 0.37).

Similarly, the mean age (±SD) and range for dogs with MPL and BPL were examined. MPL group had a mean age of 2.6 years (±2.4) and a range of 0.5 – 12 years, while BPL group had a mean age of 2.2 years (±2.2) and a range of 0.7 – 10 years. No significant difference in age at the time of surgery was observed between MPL and BPL groups (P = 0.50).

A few minor edits:

Line 64 and 65: Two sentences in a row start with “However” confusing the intended subject of the sentences.

Reply:
We appreciate your attention to the clarity of our manuscript. We have revised manuscript to eliminate the repetitive use of "however".

Line 85-86:
There is no study evaluating PLL/PL or A/PL in larger numbers of dogs with BPL in multiple small and miniature dog breeds.

Line 130: Reference needs to be filled in.

Reply:

Thank you for pointing out this oversight. We have added the missing citation. The reference list has been updated to reflect this change.

Line 189-190:
Next, we measured the PL, PLL, and vertical line (A) according to previously published methods[15].

Lines 181-183: Correct singular vs plural female(s) and male(s).

Reply:

Thank you for pointing out the inconsistency in the use of singular and plural forms. We have changed this portion to table (Table 2) to avoid confusion.

Line 246-251:
The distribution of sex and neutering status within each subgroup was described in Table 2.

Table 2. The distribution of sex and neutering status within each subgroup (n). MPL, medial patellar luxation; BPL, bidirectional patellar luxation

Intact males

Casted males

Intact females

Spayed females

MPL

Unilateral

25

4

27

9

Bilatelaral

14

1

11

4

BPL

Unilateral

10

0

4

1

Bilatelaral

7

0

2

3

MPL and Contralateral BPL

3

1

4

0

Table 1: Check consistency with headings in bold. If the bold indicates an important feature, it should be explained in the legend  

Reply:

Thank you for your comment regarding the use of bold in the headings of Table 1. Upon review, we found that the bold formatting was unintentionally applied and did not indicate any particular importance as originally intended. We have corrected this. We apologize for any confusion this may have caused and appreciate your help in improving the clarity of the manuscript.

Table 1:

MPL

BPL

Breed

Unilateral

Bilateral

Total

Unilateral

Bilateral

Total

Toy Poodle

18

2

20

14

9

23

Chihuahua

21

9

30

3

0

3

Mixed breed

6

7

13

2

3

5

Yorkshire Terrier

5

3

8

1

0

1

Pomeranian

2

2

4

2

0

2

Cavalier King Charles Spaniel

0

1

1

1

0

1

Maltese

3

3

6

0

0

0

Shibainu

6

1

7

0

0

0

Papillon

1

1

2

0

0

0

Miniature Pinscher

0

1

1

0

0

0

Pekingese

1

0

1

0

0

0

Pug

1

0

1

0

0

0

Spitz

1

0

1

0

0

0

Total

65

30

95

23

12

35

Reviewer 2 Report

Comments and Suggestions for Authors

This paper aims at characterizing bidirectional patellar luxation (BPL) in comparison with the more common condition medial patellar luxation. This work could be important in order to improve diagnostics and treatment of this condition in dogs. However, there are some concerns with the study both in statistical data handling and the way the authors attempt to infer the results.

 The study is performed based on historical clinical cases of patellar luxation in one clinic. There are no healthy controls. Still, the authors use terminology such as breed distribution, predisposition and prevalence in their title, aim in the introduction, results and discussion - using only a number of confirmed cases without controls does not support conclusions on breed distribution, predisposition or prevalence. I think the aim is better described in the abstract, where it is worded in terms of comparing radiographic measures of dogs with BPL to dogs with MPL.

 The statistical methods should be better described. To me it is rather difficult to interpret for example odds ratios when there are no controls, the authors should help the reader to interpret what the odds ratios actually mean. The data included in the study should also be revised. Performing statistics on samples of a handful of dogs per breed does not make sense. I suggest to only include Toy poodles and Chihuahuas. Possibly the other breeds could be grouped together into one larger “other breeds”-group.

I think it would be nice to include some more background on these breeds in Japan – how common are they, are there records of number of dogs for example via the Japanese Kennel Club?

In the following are a few specific comments in relation to the above, however I will not go into details until the manuscript has been re-written.

Line 1-3: Title should be re-written, the data does not support statements on breed predisposition. Something in line with “Comparison of clinical cases of bidirectional and medial patellar luxation in Toy poodles and Chihuahuas in Japan” would better describe the scope of the paper.

Line 36-39: Reads “In Japan, Toy Poodles with bidirectional patellar luxation have been frequently identified…” – lacks reference. Or is this in reference to the present work? 

Line 43-45: Repetitive.

Line 49: Reads “…(e.g., hip dysplasia)” - misuse of “e.g.”, hip dysplasia is not an example of abnormal alignement of knee and leg.

Line 54:  Reads “… as the patella exceeds” – exceed is the wrong word (means to be greater in number or size) but here the meaning should be positioned above, rewrite.

Line 63-64: Reads “… , and this proximally positioned patella was considered to be one of the anatomical causes of BPL.” Reference?

Line 68-70: Rewrite the objective to fit with what the data and study design can support.

Line 83-84: Reads “Medical records were reviewed for a thirteen-year period from 2010 to 2023 to identify dogs diagnosed with MPL who subsequently underwent surgical correction..” – only for MPL or for BPL and MPL?

Line 89-90: Reads “Age and body weight were recorded for each stifle…” – rewrite to refer to age and body weight of the dogs rather than the joints.

Line 130-131: Reads “… previously published methodsref.” – insert the missing reference.

Section 2.3: Clarify what associations that are tested.

Lines 178-184: Put in a table to make the information more readable. 

Lines 189-196, Table 2: Compare Toy poodles to Chihuahuas, these are the only breeds with sufficient number of individuals to do any statistical analysis. Remove the other breeds from the analysis, or group together into one mixed group for comparison.

Lines 201-209: Rewrite to age and body weight of dogs, not joints. Consider nesting body weight within breed, as there is potential confounding with body weight and breed.

Table 5: Why only Toy Poodles and mixed breeds?

 Line 252: Misuse of term “prevalence”, the data does not support conclusions on prevalence.

 Line 258: Reads “… especially in young and small breeds”. Should read “… especially in young dogs and small breeds”?

Comments on the Quality of English Language

The paper needs to be language reviewed before considered for publications.

Author Response

November 13, 2023

Subject: Revision and resubmission of manuscript ID vetsci-2703047to the MDPI Veterinary Sciences.

I have included the reviewer comments immediately after this letter and responded to them individually, indicating exactly how we addressed each concern or problem and describing the changes we have made. The revisions have been approved by all authors. The changes are highlighted by using the track changes mode in MS Word and using bold in the manuscript, and revised manuscript will resubmit with this letter.

Our response follows (the reviewer’s comments are in italics).

Reviewer: 2
Comments to the Author:
This paper aims at characterizing bidirectional patellar luxation (BPL) in comparison with the more common condition medial patellar luxation. This work could be important in order to improve diagnostics and treatment of this condition in dogs. However, there are some concerns with the study both in statistical data handling and the way the authors attempt to infer the results.

The study is performed based on historical clinical cases of patellar luxation in one clinic. There are no healthy controls. Still, the authors use terminology such as breed distribution, predisposition and prevalence in their title, aim in the introduction, results and discussion - using only a number of confirmed cases without controls does not support conclusions on breed distribution, predisposition or prevalence. I think the aim is better described in the abstract, where it is worded in terms of comparing radiographic measures of dogs with BPL to dogs with MPL.

Reply:

We appreciate your constructive feedback. We acknowledge that the lack of a control group in our study design limits our ability to draw conclusions regarding breed predisposition or prevalence rates for bidirectional patellar luxation (BPL). Accordingly, we have removed references to "predisposition" and "prevalence" from the manuscript, including the title, to more accurately reflect the scope and findings of our research.

We have also revised the manuscript to clarify that our comparison is not epidemiologic in nature, but rather an observational analysis of cases presented at one clinic in Japan. To this end, we focus on the relative frequency of BPL compared only to the medial patellar luxation (MPL) group, rather than making generalized claims about the breed distribution of BPL in the broader dog population. This comparison is made carefully to make it clear that it is specific to our data set and not indicative of a broader trend.

We also make a clear comparison between Toy Poodles and Chihuahuas, noting that BPL was more commonly observed in Toy Poodles within our sample, while Chihuahuas, another common breed in Japan, did not exhibit a similarly high frequency of BPL. We believe these revisions accurately represent the data and avoid any implications of a broader epidemiologic study. We appreciate the opportunity to clarify these points and improve the accuracy of the manuscript.

Line 2-4:
Bidirectional patellar luxation in small or miniature breed dogs in Japan; patient characteristics and radiographic measures compared with medial patellar luxation
Line 24-26:
The purpose of this study was to describe the patient characteristics and radiographic measures of proximodistal patellar position associated with BPL in dogs in Japan compared with dogs with medial patellar luxation (MPL).
Line 30-31:
The BPL was most commonly present in Toy Poodles (odds ratio compared to MPL dogs; 7.05) in the present study.
Line 35-36:
The study highlights the common breed for BPL in Japan, and suggests that the occurrence of BPL may be related to stifle extension.
Line 87-89:
The primary objective of this study is to describe the population characteristics including breed in dogs with BPL in Japan, and compare it with the population characteristics observed in dogs with MPL.
Line 419-420:
BPL has been observed in several breeds worldwide[1,3-5,8], but our study shows a notable presence of BPL in Toy Poodles, which accounted for 53.5% of the BPL group.
Line 423-426:
The second most common breed with BPL in our study was the Cavalier King Charles Spaniel (CKCS). However, only two CKCS were involved in the study, making it difficult to draw definitive conclusions about their predisposition to BPL.
Line 428-430:
Although there are only limited publications describing BPL and BPL is considered a relatively rare form of patellar luxation[1,3-5,8], we found a relatively high number of dogs with BPL in our study, with 35 of the 130 dogs (26.9%) exhibiting BPL.
Line 432-435:
Toy poodles are currently one of the most popular dog breeds in Japan, and their possible predisposition to BPL. These are likely to have influenced the relatively high number of dogs with BPL observed in the current study.
Line 483-485:
However, the Chihuahua, another common breed in Japan, showed a much lower presence of BPL (9.1%), resulting in a stark contrast in odds ratios for the presence of BPL compared to MPL between Toy Poodles (7.05) and Chihuahuas (0.21).
Line 496-505:
In conclusion, our investigation provides important insights into the common breed and anatomical features associated with BPL in small and miniature dog breeds in Japan. The Toy Poodle was found to be the most common breed to BPL in Japan, and the occurrence of BPL may be related to stifle extension, as patella alta in the extended stifle position was frequently observed in cases with stifle extension, although no differences in PLL/PL or A/PL ratios were found between MPL and BPL dogs.

The statistical methods should be better described. To me it is rather difficult to interpret for example odds ratios when there are no controls, the authors should help the reader to interpret what the odds ratios actually mean. The data included in the study should also be revised. Performing statistics on samples of a handful of dogs per breed does not make sense. I suggest to only include Toy poodles and Chihuahuas. Possibly the other breeds could be grouped together into one larger “other breeds”-group.

Reply:

We appreciate your feedback regarding the statistical analysis presented in our manuscript. We agree that our initial use of odds ratios may have been difficult to interpret without a control group, and that our sample size for certain breeds was too small for meaningful statistical analysis.

To address these issues, we have revised the Statistical Methods section to provide a more detailed explanation of how odds ratios were calculated and their specific interpretation in the context of our study population, which consisted of dogs diagnosed with either medial or bidirectional patellar luxation. We have clarified that these ratios do not reflect the prevalence in the general dog population, but rather the observed ratio within our clinical sample.

We have also re-evaluated the data included in our study. Following your advice, we limited our detailed statistical analysis to Toy Poodles and Chihuahuas, the breeds for which we have a sufficient number of cases. For other breeds with smaller sample sizes, we have combined them into a single "Other Breeds" category to increase the robustness of our statistical comparisons and to avoid the pitfalls of analyzing groups that are too small to yield statistically valid results. This reorganization allows for a clearer and more statistically sound comparison between groups, which we believe strengthens the findings of the paper. Thank you for your guidance in improving the statistical rigor of our study.

Line 219-222:
The odds of occurrence of BPL relative to MPL in each breed were evaluated using a two-tailed Fisher's exact test. Results were expressed as odds ratios and 95% confidence intervals (CIs). Odds ratios and 95% CIs were calculated for each breed to compare the occurrence of BPL compared to MPL.
Line 224-228:
To evaluate the association between the presence of patella alta in the extended stifle position and the type of patellar luxation (MPL or BPL), a two-tailed Fisher's exact test was performed, and the results were expressed as odds ratios comparing the occurrence of patella alta in BPL compared to MPL with their 95% CIs.
Line 256-257:
The incidence of BPL relative to MPL across various breeds, along with the corresponding odds ratios and 95% CIs were described in Table 3.
Line 259-288:
Subsequent breeds in descending order of the percent presence of BPL in dogs with MPL or BPL included the Cavalier King Charles Spaniel (50.0%), Pomeranian (33.3%), Mixed breed (27.8%), Yorkshire Terrier (11.1%), and Chihuahua (9.1%). When comparing the likelihood of developing BPL in at least one limb in the group of dogs with either MPL or BPL, Toy Poodles exhibited an odds ratio of 7.05 (95% CI: 2.82-18.6, P < 0.001), while Chihuahua had an odds ratio of 0.21 (95% CI: 0.04-0.74, P < 0.001) (Table 3). This indicates that in a group of dogs with either MPL or BPL, the incidence of BPL in Toy Poodles is 7.05 times more common than in other breeds, and the incidence of BPL in Chihuahuas is 0.21 times less common than in other breeds.
Table 3:
Presence of bidirectional patellar luxation (BPL) compared with MPL in dogs in Japan. This table presents the odds ratios for the presence of BPL compared with MPL in various dog breeds, analyzed using a two-tailed Fisher's exact test. Breeds with a small number of cases were combined into a group “other breeds” for statistical analysis. MPL, medial patellar luxation; BPL, bidirectional patellar luxation; CI, confidence interval.

Breed

Total (n)

The number of dogs with  MPL (n)

The number of dogs with BPL (n)

Occurrence of BPL (%)

Odds ratio

Lower 95% CI

Upper 95% CI

P value

Toy Poodle

43

20

23

53.5

7.05

2.82

18.6

<0.001

Chihuahua

33

30

3

9.1

0.21

0.04

0.74

<0.001

Other breeds

Mix

18

13

5

27.8

0.39

0.14

0.97

<0.05

Yorkshire Terrier

9

8

1

11.1

Shibainu

7

7

0

0

Pomeranian

6

4

2

33.3

Maltese

6

6

0

0

Cavalier King Charles Spaniel

2

1

1

50.0

Papillon

2

2

0

0

Miniature Pinscher

1

1

0

0

Pekingese

1

1

0

0

Pug

1

1

0

0

Spitz

1

1

0

0

I think it would be nice to include some more background on these breeds in Japan – how common are they, are there records of number of dogs for example via the Japanese Kennel Club?

Reply:

Thank you for your suggestion to include background information on the breeds included in our study. We have now added data on the popularity of Toy Poodles in Japan to the manuscript, based on records from the Japan Kennel Club. According to their database, the Toy Poodle was indeed the most popular breed registered during our study period from 2010 to 2023. This addition is supported by appropriate references to published data from the Japan Kennel Club, providing a reliable snapshot of breed popularity over time and further grounding our study in its specific geographic and cultural context.

Line 420-423:
This prevalence could be due to the actual breed predisposition to BPL, the popularity of Toy Poodles in Japan[27,28]; Toy poodle was most common breed in Japan in the study period (2010-2023) reported by Japan Kennel Club[29,30], or a combination of both.

In the following are a few specific comments in relation to the above, however I will not go into details until the manuscript has been re-written.

Line 1-3: Title should be re-written, the data does not support statements on breed predisposition. Something in line with “Comparison of clinical cases of bidirectional and medial patellar luxation in Toy poodles and Chihuahuas in Japan” would better describe the scope of the paper.

Reply:

In response to your recommendation, we have carefully revised the title to more accurately reflect the data and scope of our research. The new title is now "Bidirectional patellar luxation in small or miniature breed dogs in Japan; patient characteristics and radiographic measures compared with medial patellar luxation," which avoids implying a broader breed predisposition. This title is consistent with the content of our manuscript and directly addresses your concerns regarding the presentation of the results of our study. Thank you for guiding us to be more precise in our wording.

Line 2-4:
Bidirectional patellar luxation in small or miniature breed dogs in Japan; patient characteristics and radiographic measures compared with medial patellar luxation

Line 36-39: Reads “In Japan, Toy Poodles with bidirectional patellar luxation have been frequently identified…” – lacks reference. Or is this in reference to the present work? 

Reply:

We appreciate your request for clarification regarding the statement on Toy Poodles with bidirectional patellar luxation (BPL) in Japan. The observation cited was indeed a novel finding from our current study and not previously documented in the literature, which might explain the lack of references in the original manuscript. We have now revised the sentence to clearly reflect that it presents our own research findings. Additionally, we have cited a publication that discusses the incidence of BPL in Toy Poodles in Japan to provide context and demonstrate the scarcity of existing research on this specific topic. Our updated text is as follows.

Line 59-62:
Although the authors have observed the BPL more frequently in Toy Poodles in Japan, to date, to the best of the author's knowledge, only one publication in Japanese has been published describing the incidence of BPL in Toy Poodles in Japan[7]. No previous studies have investigated common breeds with BPL in small or miniature breeds in Japan.

Line 43-45: Repetitive.

Reply:

Thank you for pointing out the repetitive elements in lines 43-45 of our manuscript. We have removed the redundancy to improve the flow and conciseness of the text.

Line 49: Reads “…(e.g., hip dysplasia)” - misuse of “e.g.”, hip dysplasia is not an example of abnormal alignement of knee and leg.

Reply:

We appreciate your attention to the misuse of "e.g." in line 49. Upon review, we recognize that hip dysplasia was incorrectly used as an example of abnormal alignment of the knee and leg. We have removed this example. We have also added a new reference.

Line 69-71:
These features include a shallow or absent femoral groove, abnormal alignment of the knee and leg, skeletal abnormalities, or hormonal influences during growth, particularly in small breed dogs with medial patellar luxation (MPL)[9-11].

Line 54:  Reads “… as the patella exceeds” – exceed is the wrong word (means to be greater in number or size) but here the meaning should be positioned above, rewrite.

Reply:

Thank you for pointing out the incorrect use of "exceeds" in line 54. We have revised the sentence to accurately convey the intended meaning.

Line 74-76:
A proximally positioned patella (patella alta) has been reported to be associated with patellar instability in humans[17-20], as the patella positioned proximal to the femoral trochlea and loses support from the trochlear ridge. 

Line 63-64: Reads “… , and this proximally positioned patella was considered to be one of the anatomical causes of BPL.” Reference?

Reply:

Thank you for requesting a reference for the statement on line 63-64 regarding the anatomical causes of BPL. After reviewing the literature, we found that the specific relationship between a proximally positioned patella and BPL is not conclusively supported by the existing literature. We noted that a previous study measuring the proximodistal patella position radiographically in Pomeranians with BPL did not find significant differences compared to healthy dogs. Given the lack of robust evidence to support this theory, we chose to remove this statement from our manuscript to maintain the scientific accuracy and integrity of our discussion.

Line 82-84:
During this overextension, the patella is positioned proximally in the trochlear groove, resulting in reduced support from both the medial and lateral trochlear ridges.

Line 68-70: Rewrite the objective to fit with what the data and study design can support.

Reply:

Thank you for your guidance to ensure that our study objectives are consistent with what the data and study design can support. Based on your guidance, we have revised the objective to accurately reflect the scope of our research. This revised objective is carefully aligned with the data we have collected and avoids implications of breed predisposition or prevalence that our study design does not support. We trust that this revision solidifies the focus and intent of our research.

Line 87-89:
The primary objective of this study is to describe the population characteristics including breed in dogs with BPL in Japan, and compare it with the population characteristics observed in dogs with MPL.

Line 83-84: Reads “Medical records were reviewed for a thirteen-year period from 2010 to 2023 to identify dogs diagnosed with MPL who subsequently underwent surgical correction..” – only for MPL or for BPL and MPL?

Reply:

We appreciate your request for clarification. Our study does indeed include dogs with both medial patellar luxation (MPL) and bidirectional patellar luxation (BPL). Initially, we considered BPL to be a subset of MPL, which led to the broad term "MPL" encompassing both conditions in our inclusion criteria. Recognizing the potential for confusion, we have now explicitly separated MPL and BPL in our manuscript. The revised lines specify that the medical record review from 2010 to 2023 included dogs diagnosed with both MPL and BPL that underwent surgical correction. This change ensures that our inclusion criteria are transparent and accurately reflect the scope of our study. Thank you for your guidance on this clarification.

Line 101-102:
Medical records were reviewed for a thirteen-year period from 2010 to 2023 to identify dogs diagnosed with MPL or BPL who subsequently underwent surgical correction.

Line 89-90: Reads “Age and body weight were recorded for each stifle…” – rewrite to refer to age and body weight of the dogs rather than the joints.

Reply:

We acknowledge the confusion caused by the original wording. To clarify, we have revised the sentence to accurately reflect that the age and body weight recorded are attributes of the dogs, not their joints. Thank you for helping us improve the accuracy of the manuscript.

Line 107-108:
Age and body weight of dogs were recorded for each stifle at the time of surgery for MPL or BPL.
Line 289-298:
The mean body weight (±SD) and range for dogs with MPL and BPL were as follows: MPL group had a mean weight of 4.01 kg (±2.43.) and a range of 1.7 – 15 kg, while BPL group had a mean weight of 3.34 kg (±1.26) and a range of 1.3 – 7.0 kg. No significant difference in body weight at the time of surgery was observed between MPL and BPL groups (P = 0.37).
Similarly, the mean age (±SD) and range for dogs with MPL and BPL were examined. MPL group had a mean age of 2.6 years (±2.4) and a range of 0.5 – 12 years, while BPL group had a mean age of 2.2 years (±2.2) and a range of 0.7 – 10 years. No significant difference in age at the time of surgery was observed between MPL and BPL groups (P = 0.50).

Line 130-131: Reads “… previously published methodsref.” – insert the missing reference.

Reply:

Thank you for pointing out the oversight of the missing reference to the methods used in our study, as mentioned in lines 130-131. We have now added the appropriate citation. We apologize for the initial omission and appreciate your help in correcting it.

Line 189-190:
Next, we measured the PL, PLL, and vertical line (A) according to previously published methods[15].

Section 2.3: Clarify what associations that are tested.

Reply:

Thank you for your valuable feedback on Section 2.3. We have now expanded the section to explicitly describe the associations examined in our analysis. We have included a more thorough explanation of how the odds ratios were calculated and interpreted in the context of these associations. We appreciate the opportunity to refine our manuscript accordingly.

Line 219-222:
The odds of occurrence of BPL relative to MPL in each breed were evaluated using a two-tailed Fisher's exact test. Results were expressed as odds ratios and 95% confidence intervals (CIs). Odds ratios and 95% CIs were calculated for each breed to compare the occurrence of BPL compared to MPL.
Line 224-228:
To evaluate the association between the presence of patella alta in the extended stifle position and the type of patellar luxation (MPL or BPL), a two-tailed Fisher's exact test was performed, and the results were expressed as odds ratios comparing the occurrence of patella alta in BPL compared to MPL with their 95% CIs.

Lines 178-184: Put in a table to make the information more readable. 

Reply:

Thank you for recommending that we present the information in lines 178-184 in a table format. We have created a table that effectively organizes this data, making it more accessible and easier for the reader to interpret.

Line 246-249:

The distribution of sex and neutering status within each subgroup was described in Table 2.

Table 2. The distribution of sex and neutering status within each subgroup (n). MPL, medial patellar luxation; BPL, bidirectional patellar luxation

Intact males

Casted males

Intact females

Spayed females

MPL

Unilateral

25

4

27

9

Bilatelaral

14

1

11

4

BPL

Unilateral

10

0

4

1

Bilatelaral

7

0

2

3

MPL and Contralateral BPL

3

1

4

0

Lines 189-196, Table 2: Compare Toy poodles to Chihuahuas, these are the only breeds with sufficient number of individuals to do any statistical analysis. Remove the other breeds from the analysis, or group together into one mixed group for comparison.

Reply:

In accordance with your recommendation, we have revised Table 2 (now Table 3) to focus our statistical analysis on Toy Poodles and Chihuahuas, recognizing that these breeds have an adequate sample size to allow for meaningful analysis. We have combined the data from other breeds with smaller sample sizes into a collective "Other breeds" category. This approach enhances the statistical integrity of the comparisons and addresses the potential for error due to small group sizes. We appreciate your suggestions for improving the study's statistical foundation.

Table 3:
Presence of bidirectional patellar luxation (BPL) compared with MPL in dogs in Japan. This table presents the odds ratios for the presence of BPL compared with MPL in various dog breeds, analyzed using a two-tailed Fisher's exact test. Breeds with a small number of cases were combined into a group “other breeds” for statistical analysis. MPL, medial patellar luxation; BPL, bidirectional patellar luxation; CI, confidence interval.

Breed

Total (n)

The number of dogs with  MPL (n)

The number of dogs with BPL (n)

Occurrence of BPL (%)

Odds ratio

Lower 95% CI

Upper 95% CI

P value

Toy Poodle

43

20

23

53.5

7.05

2.82

18.6

<0.001

Chihuahua

33

30

3

9.1

0.21

0.04

0.74

<0.001

Other breeds

Mix

18

13

5

27.8

0.39

0.14

0.97

<0.05

Yorkshire Terrier

9

8

1

11.1

Shibainu

7

7

0

0

Pomeranian

6

4

2

33.3

Maltese

6

6

0

0

Cavalier King Charles Spaniel

2

1

1

50.0

Papillon

2

2

0

0

Miniature Pinscher

1

1

0

0

Pekingese

1

1

0

0

Pug

1

1

0

0

Spitz

1

1

0

0

Lines 201-209: Rewrite to age and body weight of dogs, not joints. Consider nesting body weight within breed, as there is potential confounding with body weight and breed.

Reply:

We have amended the manuscript to correctly attribute age and body weight to the dogs themselves, rather than to their joints, as per your guidance. Regarding the suggestion to nest body weight within breed to account for potential confounding, we have reviewed our data set and determined that our current sample size does not provide sufficient diversity to robustly perform such a nested analysis. We appreciate your recommendations, which have resulted in a more thorough and nuanced presentation of our study.

Line 108-109:
Age and body weight of dogs were recorded for each stifle at the time of surgery for MPL or BPL.
Line 290-299:
The mean body weight (±SD) and range for dogs with MPL and BPL were as follows: MPL group had a mean weight of 4.01 kg (±2.43.) and a range of 1.7 – 15 kg, while BPL group had a mean weight of 3.34 kg (±1.26) and a range of 1.3 – 7.0 kg. No significant difference in body weight at the time of surgery was observed between MPL and BPL groups (P = 0.37).
Similarly, the mean age (±SD) and range for dogs with MPL and BPL were examined. MPL group had a mean age of 2.6 years (±2.4) and a range of 0.5 – 12 years, while BPL group had a mean age of 2.2 years (±2.2) and a range of 0.7 – 10 years. No significant difference in age at the time of surgery was observed between MPL and BPL groups (P = 0.50).

Table 5: Why only Toy Poodles and mixed breeds?

Reply:

In response to your question about the focus on Toy Poodles and Mixed breeds in Table 5, we have clarified in the manuscript that these groups were selected for statistical analysis based on adequate sample sizes. Other breeds in our study did not have a sufficient number of cases to ensure the validity of a Student's T-test. We have included this explanation in the manuscript to inform readers of the rationale behind our selection of breeds for statistical comparison and to be transparent about the limitations of our data set.

Line 364-366:
Breeds other than Toy Poodle and Mixed breed were not included in Table 6 because there were not enough numbers in one or both groups to perform statistical analysis.

Line 252: Misuse of term “prevalence”, the data does not support conclusions on prevalence.

Reply:

We acknowledge your comment regarding the misuse of the term "prevalence" in line 252. We have removed this term and rephrased the sentence to accurately reflect the nature of our data, which do not allow epidemiologic conclusions about prevalence rates.

Line 389-391:
Our study aimed to describe the common breed and anatomical features associated with BPL in small and miniature dog breeds in Japan, in comparison with MPL. We found that the Toy Poodle breed was the most commonly diagnosed with BPL, aligning with our initial hypothesis. Furthermore, the presence of BPL compared to MPL was higher in our study than generally reported, which could be due to the popularity of the Toy Poodle breed in Japan.

Line 258: Reads “… especially in young and small breeds”. Should read “… especially in young dogs and small breeds”?

Reply:

We appreciate your attention to detail on line 258. The phrase has been corrected to read "...especially in young dogs and small breeds".

Line 395-396:
MPL is a common orthopedic problem in dogs, especially in young dogs and small breeds[1,25,26].

Round 2

Reviewer 2 Report

Comments and Suggestions for Authors

All my comments on the first version of the manuscript have been addressed, and I am for the most part happy with the changes and that the title, objective and discussion now more accurately reflect the data and study design. 

Some minor comments: 

Line 36, 268, 366: "the common breed" - this has been added to replace breed predisposition, but I don't think it is clear what it means and I don't think it is proper English. I am not quite clear on the intended meaning, but I suggest: 

Line 36: "The study highlights the common nbreed for BPL in Japan, ..". Suggest "The study highlights BPL in different dog breeds in Japan, ...". 

Line 268: "Our study aimed to describe the common breed and anatomical features associated with BPL in small and miniature dog breeds in Japan, .."
Suggest: "Our study aimed to describe anatomical features associated with BPL in small and miniature dog breeds in Japan, .."

Line 366: "In conclusion our investigation provides important insights into the common breed and anatomical features associated with BPL in small and miniature dog breeds in Japan" 
Suggest: "In conclusion our investigation provides important insights into anatomical features and breed characteristics associated with BPL in small and miniature dog breeds in Japan". 

Line 268, conclusions: Reads "The Toy Poodle was found to be the most common breed to BPL in Japan". Rewrite to reflect that Toy Poodle was the most common breed to be surgically treated for BPL in this study - I don't think the authors can expand their results to be valid for all clinics in Japan. 

Table 2: "Casted" should read castrated

Table 3: It seems as if the same table appears has been copied in twice. To avoid misunderstanding the "other breeds" category should be summarised into one line. The information about numbers per breed is available in Table 1. Summarize all the breeds into one row and present odds ratios. As it is now it becomes confusing if the odds ratios given refers only to Cavaliers (in the first copy of the table, Pomeranians in the second). 

Lines 252-266, including Table 6: I would suggest to remove this, and replace with at sentence, something in line with "We also evaluated the ratios of PLL/PL and A/PL within the two breed groups with sufficient number of observations (Toy Poodle and Mixed breeds) but found no significant differences. In any case Line 252-256 duplicates with Line 257-260 I think.  

Comments on the Quality of English Language

With some minor corrections as commented above I think it is acceptable, but I would also recommend to have the manuscript read through by someone native to the English language to further improve the readability. 

Author Response

November 29, 2023

Subject: Revision and resubmission of manuscript ID vetsci-2703047 to the MDPI Veterinary Sciences.

I have included the reviewer comments immediately after this letter and responded to them individually, indicating exactly how we addressed each concern or problem and describing the changes we have made. The revisions have been approved by all authors. The changes are highlighted by using the track changes mode in MS Word and using bold in the manuscript, and revised manuscript will resubmit with this letter.

Our response follows (the reviewer’s comments are in italics).

Reviewer: 1

All my comments on the first version of the manuscript have been addressed, and I am for the most part happy with the changes and that the title, objective and discussion now more accurately reflect the data and study design. 

Some minor comments: 

Line 36, 268, 366: "the common breed" - this has been added to replace breed predisposition, but I don't think it is clear what it means and I don't think it is proper English. I am not quite clear on the intended meaning, but I suggest: 

Line 36: "The study highlights the common nbreed for BPL in Japan, ..". Suggest "The study highlights BPL in different dog breeds in Japan, ...". 

Line 268: "Our study aimed to describe the common breed and anatomical features associated with BPL in small and miniature dog breeds in Japan, .."
Suggest: "Our study aimed to describe anatomical features associated with BPL in small and miniature dog breeds in Japan, .."

Line 366: "In conclusion our investigation provides important insights into the common breed and anatomical features associated with BPL in small and miniature dog breeds in Japan" 
Suggest: "In conclusion our investigation provides important insights into anatomical features and breed characteristics associated with BPL in small and miniature dog breeds in Japan". 

Reply:
Thank you for your constructive comments. We have carefully reviewed your suggestions and have revised the manuscript accordingly. Specifically, we have removed the term "common breed" to avoid ambiguity and to ensure clarity in the text.

Line 36-37:
The study highlights BPL in different dog breeds in Japan, and suggests that the occurrence of BPL may be related to stifle extension.
Line 265-266:
Our study aimed to describe anatomical features associated with BPL in small and miniature dog breeds in Japan, in comparison with MPL.

Line 381-382:
In conclusion, our investigation provides important insights into anatomical features and breed characteristics associated with BPL in small and miniature dog breeds in Japan.

Line 268, conclusions: Reads "The Toy Poodle was found to be the most common breed to BPL in Japan". Rewrite to reflect that Toy Poodle was the most common breed to be surgically treated for BPL in this study - I don't think the authors can expand their results to be valid for all clinics in Japan. 

Reply:
Thank you for pointing out the need for precision in describing the conclusion of our findings. We have revised the text to accurately reflect that the Toy Poodle was the most common breed surgically treated for BPL within this study. We agree that our results cannot be generalized to all clinics in Japan without further research.

Line 383-386:
The Toy Poodle was the most common breed to be surgically treated for BPL in this study, and the occurrence of BPL may be related to stifle extension, as patella alta in the extended stifle position was frequently observed in cases with stifle extension, although no differences in PLL/PL or A/PL ratios were found between MPL and BPL dogs.

Table 2: "Casted" should read castrated

Reply:
Your attention to detail is appreciated. The typographical error in Table 2 has been corrected to read "castrated". Thank you for bringing this to our attention.

Intact males

Castrated males

Intact females

Spayed females

MPL

Unilateral

25

4

27

9

Bilatelaral

14

1

11

4

BPL

Unilateral

10

0

4

1

Bilatelaral

7

0

2

3

MPL and Contralateral BPL

3

1

4

0

Table 2:

Table 3: It seems as if the same table appears has been copied in twice. To avoid misunderstanding the "other breeds" category should be summarised into one line. The information about numbers per breed is available in Table 1. Summarize all the breeds into one row and present odds ratios. As it is now it becomes confusing if the odds ratios given refers only to Cavaliers (in the first copy of the table, Pomeranians in the second). 

Reply:
We appreciate your comment regarding the duplication in Table 3. We have corrected this error by removing the duplicate item and combining the "other breeds" category into a single row, as suggested. Thank you for helping us improve the clarity and accuracy of our data presentation.

Table 3:

Breed

Total

The number of dogs with MPL

The number of dogs with BPL

Occurrence of BPL (%)

Odds ratio

Lower 95% CI

Upper 95% CI

P value

Toy Poodle

43

20

23

53.5

7.05

2.82

18.6

<0.001

Chihuahua

33

30

3

9.1

0.21

0.04

0.74

<0.001

Other breeds

54

45

9

16.7

0.39

0.14

0.97

<0.05

Lines 252-266, including Table 6: I would suggest to remove this, and replace with at sentence, something in line with "We also evaluated the ratios of PLL/PL and A/PL within the two breed groups with sufficient number of observations (Toy Poodle and Mixed breeds) but found no significant differences. In any case Line 252-256 duplicates with Line 257-260 I think.  

Reply:
Thank you for your insightful suggestion regarding the content of lines 252-266 and Table 6. Following your recommendation, we have removed the table and replaced it with a concise summary sentence in the manuscript. We have also corrected the duplication between lines 252-256 and 257-260. We appreciate your guidance in improving the clarity and conciseness of our manuscript.

Line 259-261:
In the present study, we also evaluated the ratios of PLL/PL and A/PL within the two breed groups with sufficient number of observations (Toy Poodle and Mixed breed) but found no significant differences.
